# Systemic Impact of Platelet Activation in Abdominal Surgery: From Oxidative and Inflammatory Pathways to Postoperative Complications

**DOI:** 10.3390/ijms26157150

**Published:** 2025-07-24

**Authors:** Dragos-Viorel Scripcariu, Bogdan Huzum, Cornelia Mircea, Dragos-Florin Tesoi, Oana-Viola Badulescu

**Affiliations:** 1Department of Surgical Sciences, Faculty of Medicine, University of Medicine and Pharmacy “Grigore T. Popa”, 700115 Iasi, Romania; dragos-viorel.scripcariu@umfiasi.ro; 2Faculty of Pharmacy, University of Medicine and Pharmacy “Grigore T. Popa”, 700115 Iasi, Romania; cornelia.mircea@umfiasi.ro; 3Department of Pediatry, University of Medicine and Pharmacy “Grigore T. Popa”, 700115 Iasi, Romania; dragos-florin.tesoi@email.umfiasi.ro; 4Department of Pathophysiology, Faculty of Medicine, University of Medicine and Pharmacy “Grigore T. Popa”, 700115 Iasi, Romania; oana.badulescu@umfiasi.ro

**Keywords:** platelets, oxidative stress, inflammation, abdominal surgery, biomarkers, personalized medicine

## Abstract

Although platelets have been traditionally thought of to be essential hemostasis mediators, new research shows how important they are for controlling cellular oxidative stress, inflammatory processes, and immunological responses—particularly during major surgery on the abdomen. Perioperative problems are largely caused by the continually changing interaction of inflammatory cytokines, the formation of reactive oxygen species (ROS), and platelet activation. The purpose of this review is to summarize the most recent data regarding the complex function of platelets in abdominal surgery, with an emphasis on how they interact with inflammation and oxidative stress, and to investigate the impact on postoperative therapy and subsequent studies. Recent study data on platelet biology, redox signals, surgical stress, and antiplatelet tactics was reviewed in a systematic manner. Novel tailored therapies, perioperative antiplatelet medication, oxidative biomarkers of interest, and platelet-derived microscopic particles are important themes. In surgical procedures, oxidative stress dramatically increases the reactive capacity of platelets, spurring thromboinflammatory processes that affect cardiac attacks, infection risk, and recovery. A number of biomarkers, including soluble CD40L, thromboxane B2, and sNOX2-derived peptide, showed potential in forecasting results and tailored treatment. Antiplatelet medications are still essential for controlling risk factors for cardiovascular disease, yet using them during surgery necessitates carefully weighing the risks of thrombosis and bleeding. Biomarker-guided therapies, antioxidant adjuncts, and specific platelet inhibitors are examples of evolving tactics. In abdominal procedures, platelets strategically operate at the nexus of oxidative stress, inflammatory processes, and clotting. Improved patient classification, fewer problems, and the creation of individualized surgical care strategies could result from an increased incorporation of platelet-focused tests and therapies into perioperative processes. To improve clinical recommendations, subsequent studies may want to focus on randomized studies, biomarker verification, and using translational approaches.

## 1. Introduction

Severe physiological strain as well as subsequent challenges are frequently linked to abdominal surgery, which can range from planned therapies to emergencies. The possibility of unfavorable consequences such thromboembolic events, systemic inflammation, delayed wound healing, and oxidative tissue damage persists as a significant therapeutic concern despite improvements in surgical methods and perioperative care [1,2]. Platelet stimulation, which is essential for hemostasis as well as inflammatory conditions, immunology, and redox signaling, is just one of the main factors influencing this reaction. Platelets were once thought to be only involved in the production of clots, but they are now understood to be active regulators of vascular homeostasis and immunological responses [3,4]. They quickly release granules, express surface adhesion molecules, and shed platelet-derived microparticles (PMPs) in response to endothelial damage and oxidative stress that occurs during operating trauma. These actions then trigger inflammatory and hemorrhaging chain reactions.

The clinical significance of platelet parameters in surgical procedures has been brought to light by recent investigations. It has been demonstrated that the degree of inflammatory processes, edema, and surgical discomfort are correlated with perioperative shifts in platelet indicators, such as plateletcrit, mean platelet volume (MPV), and platelet distribution width (PDW), especially in procedures requiring substantial tissue handling [3]. While platelet parameters like MPV and PDW are readily available and have shown potential as biomarkers in some studies (particularly in specific contexts like acute coronary syndromes or neonatal care), their clinical utility in predicting outcomes across various conditions still requires further investigation [5]. More rigorous, large-scale studies are necessary to determine their true predictive value and to establish clear cut-off values for different clinical contexts.

Furthermore, the perioperative platelet count is becoming more widely acknowledged as a predictive biomarker for issues that may arise after surgery, such as infection, thrombosis, and extended hospitalization [6]. Furthermore, platelet-derived microparticles (PMPs) have become important mediators at the coagulation–inflammatory interface. After surgery, endothelial cell dysfunction and widespread thromboinflammation are exacerbated by these submicron vesicles, which are produced from active platelets and express phosphatidylserine along with a variety of proinflammatory and procoagulant factors [4].

With an emphasis on their role in oxidative stress, inflammation, and postoperative problems, this review attempts to examine the complex role of platelets in abdominal surgical procedures. In addition to highlighting the potential of platelet-related indicators and therapeutic targets in enhancing perioperative care, we summarize current mechanistic and clinical data on platelet activation and its significance for patient outcomes.

## 2. Platelet Biology and Activation Mechanisms

Megakaryocytes produce platelets, which are tiny, anucleate cell fragments that are well-known for their crucial function in hemostasis. But in the last ten years, more and more data have shown that they actively participate in immunological responses, inflammatory signaling, and redox control, especially when tissue damage and surgical trauma are involved [7,8]. The quick activation of circulating platelets is a critical factor in determining healing and postoperative problems during abdominal surgery, where tissue injury and endothelium disruption are unavoidable. Dense granules containing ADP, serotonin, and calcium ions are seen in resting platelets, which have a diameter of 2–4 μm and contain α-granules rich in P-selectin, platelet-derived growth factor (PDGF), and von Willebrand factor (vWF). Platelet mitochondria are essential for redox balance and bioenergetics [9]. Prostacyclin (PGI_2_) and endothelial-derived nitric oxide (NO) keep platelets dormant under normal settings. Glycoprotein receptors like GPVI and integrin α2β1 mediate platelet adherence to damaged subendothelial matrix elements, particularly collagen, after vascular damage (Figure 1). Granule discharge, integrin activation, and platelet aggregation are the outcomes of a series of intracellular signaling processes that are started by this contact [10]. Bidirectional signaling, involving both inside-out and outside-in signaling pathways, is crucial for regulating platelet adhesion, spreading, and stable aggregation during hemostasis and thrombosis. Inside-out signaling activates integrins, particularly αIIbβ3, on the platelet surface, increasing their affinity for adhesive proteins such as fibrinogen. This allows platelets to bind to the damaged vessel wall. Binding of adhesive proteins to activated integrins triggers outside-in signaling. This signaling cascade initiates a series of cellular events, including platelet spreading, cytoskeletal reorganization, granule secretion, and the stabilization of platelet adhesion and aggregation, contributing to clot formation [11].

P-selectin is a cell adhesion molecule found on the surface of activated platelets and endothelial cells. The major ligand for P-selectin is PSGL-1, which is expressed on the surface of leukocytes. When platelets are activated (e.g., by injury or inflammation), they express P-selectin, which then binds to PSGL-1 on circulating leukocytes. This initial interaction causes leukocytes to tether and roll along the activated platelets. The interaction between P-selectin on activated platelets and PSGL-1 on leukocytes is a key mechanism in the formation of platelet–leukocyte aggregates, which are major contributors to thromboinflammation. This interaction facilitates the recruitment of leukocytes to sites of inflammation and injury, where they can contribute to clot formation and the inflammatory response. The P-selectin/PSGL-1 pathway is a potential therapeutic target for preventing or treating thromboinflammatory conditions. Inhibiting this interaction could reduce the formation of platelet–leukocyte aggregates and mitigate the associated inflammatory and thrombotic events [12].

Platelet subpopulations exhibit functional differences based on their age. Younger platelets are more responsive to stimuli, readily adhering to damaged tissue and aggregating to form clots. They also have more mitochondria, which are crucial for their energy production and function. Moreover, younger platelets are more involved in the early stages of hemostasis, promoting clot formation and stability, while they tend to have a less prominent role in inflammatory processes compared to older platelets. The latter may show a decreased ability to adhere, aggregate, and respond to stimuli and may have lower mitochondrial content and reduced mitochondrial function. While still involved in hemostasis, older platelets may contribute more to inflammatory responses and interactions with leukocytes. More research is needed to identify specific biomarkers for different platelet subpopulations and their roles in various types of surgical interventions [13].

Several convergent signaling channels coordinate platelet activation. G-protein–coupled receptor (GPCR) pathways, in which ADP binds to P2Y_1_ and P2Y_12_ receptors, are some of the most significant. GPCRs play a crucial role in platelet activation and function, particularly in response to various agonists involved in hemostasis and thrombosis. These receptors, upon binding to agonists such as ADP, thromboxane A2, and thrombin, trigger intracellular signaling pathways that lead to platelet activation, aggregation, and ultimately, thrombus formation [14,15]. While P2Y_12_ promotes aggregation by stimulating PI3K/Akt and regulating the GPIIb/IIIa integrin complex, the P2Y_1_ receptor triggers platelet shape shift and calcium influx through phospholipase C stimulation [16]. At the same time, platelet mobilization and degranulation are enhanced by thromboxane A_2_ (TxA_2_), which is produced through the cyclooxygenase-1 (COX-1) pathway. To improve the responsiveness of platelets and integrin stimulation, other receptors like PAR1 (activated by thrombin) and CLEC-2 activate tyrosine kinase-dependent pathways (Syk, PLCγ2, and MAPK) [17]. PAR1 is activated by the proteolytic cleavage of its extracellular N-terminal domain by enzymes such as thrombin. This cleavage exposes a new N-terminal sequence that acts as a tethered ligand, binding to the receptor and triggering intracellular signaling that leads to platelet activation, aggregation, and ultimately clot formation [18]. The CLEC-2 (C-type lectin-like receptor 2) pathway is a signaling cascade in platelets that is triggered by the binding of specific ligands, such as podoplanin or rhodocytin, to the CLEC-2 receptor on the platelet surface. This binding initiates a series of intracellular events that ultimately lead to platelet activation and aggregation [19,20].

A subset of highly activated platelets changes into a procoagulant phenotype, which is defined by externalizing of phosphatidylserine (PS) and phosphatidylethanolamine, prolonged intracellular calcium increase, and mitochondrial depolarization. This phenotype unexpectedly inhibits aggregation through GPIIb/IIIa deactivation, while simultaneously promoting thrombin production and supporting the assembly of coagulation complexes (tenase and prothrombinase) [17]. These platelets aid in the creation of fibrin networks, especially in the center of thrombi that form at the trauma site. Via redox-sensitive signals, reactive oxygen species (ROS) further alter the stimulation of platelets. Platelets produce ROS through NADPH oxidase (NOX) and mitochondrial metabolism when stimulated by thrombin or collagen. These ROS maintain aggregation and facilitate the procoagulant transition by increasing intracellular calcium discharge, granule secretion, and GPIIb/IIIa activation [21]. ROS are important secondary messengers, but too much of them can be harmful. For this reason, endogenous antioxidant mechanisms, including glutathione peroxidase, catalase, and superoxide dismutase (SOD), maintain platelets’ redox balance [9]. Following stimulation, platelets release extracellular vesicles called platelet-derived microparticles (PMPs) in addition to soluble mediators. These microparticles play a major role in endothelial activation and thromboinflammation by exposing PS and carrying proteins like CD40 ligand. The bulk of circulating microparticles are PMPs, which have the ability to spread inflammatory signals to far-off locations. Increased postoperative complications and a prothrombotic condition have been linked to greater PMP levels in surgical settings [4,22]. The dynamic interaction between activated platelets, leukocytes, and endothelial cells is a critical process in inflammation. Surgery can induce endothelial dysfunction, leading to increased expression of adhesion molecules like ICAM-1 (Intercellular Adhesion Molecule-1) and VCAM-1 (Vascular Cell Adhesion Molecule-1) on endothelial cells. ICAM-1, a cell surface glycoprotein, facilitates the adhesion of leukocytes to endothelial cells via its binding to integrins. It is particularly important in neutrophil and monocyte recruitment. VCAM-1 also plays a significant role in leukocyte adhesion, especially monocytes, to the endothelium. In addition, platelet CD40L (a protein on the platelet surface) can bind to endothelial CD40, leading to increased expression of ICAM-1 and VCAM-1 on endothelial cells. It has been demonstrated that interaction between platelets and endothelial cells, facilitated by adhesion molecules, promotes the attraction of leukocytes to sites of inflammation. Platelet activation can also lead to the shedding of platelet-derived microparticles, which can further modulate endothelial cell function and leukocyte recruitment [23,24]. Together, mechanical, inflammatory, and redox stressors all contribute to platelet activation after abdominal surgery. Together, these triggers change platelets from passive circulators into key players in vascular function, immunological signaling, and coagulation regulation. Finding new treatment targets and improving perioperative care for patients undergoing surgery require an increased awareness of such pathways.

Platelet production and turnover, primarily regulated by thrombopoietin (TPO), can significantly influence perioperative platelet function and recovery. TPO, synthesized by the liver and kidneys, stimulates the bone marrow to produce megakaryocytes, which then mature into platelets [25]. Patients with pre-existing thrombocytopenia may have impaired platelet function, increasing the risk of bleeding during and after surgery.

Conversely, thrombocytosis can also pose risks. While it may seem protective, it can also lead to increased platelet activation and aggregation, potentially contributing to thrombotic complications. Inflammation, common in surgical settings, can affect platelet function and turnover. Cytokines released during inflammation can stimulate platelet production and activation, potentially leading to both bleeding and thrombotic complications. Following hepatic surgery, particularly major resections, a temporary decrease in thrombopoietin levels often occurs, especially in the early postoperative period. This reduction, primarily due to the liver being the major site of TPO production, can contribute to thrombocytopenia. However, platelet counts typically recover as TPO levels rise, and the extent of platelet count decrease may not differ significantly between major and minor hepatic resections, despite the larger TPO reduction after major surgery [26]. Monitoring platelet production and turnover markers like TPO levels, platelet size (mean platelet volume), and the proportion of reticulated platelets can provide insights into a patient’s platelet dynamics and help predict perioperative risks. Thrombopoietin receptor agonists are a class of drugs used to treat thrombocytopenia, particularly in perioperative settings. They are often used as an alternative to platelet transfusions, especially in patients with chronic liver disease or other conditions where platelet transfusions carry increased risks [27,28,29].

## 3. Surgical Stress and Systemic Inflammation

Enhanced release of catecholamines, cortisol, and pro-inflammatory cytokines like IL 1, IL 6, and TNF α is indicative of a widespread reaction to stress brought on by abdominal surgery, which is marked by stimulation of the neuroendocrine and immunological systems [30]. Although necessary for managing tissue damage, this concerted reaction can become imbalanced and increase the risk of perioperative complications. One characteristic of this reaction is oxidative stress. Reactive oxygen species (ROS) buildup and antioxidant reserves are depleted during major abdominal procedures such hepatectomies or colorectal resections. This damages endothelial cells and worsens inflammation [2,31]. For example, a postoperative decrease in antioxidant capacity was significantly linked to severe morbidity in patients who had liver resection (*p* = 0.017) [32].

Low perioperative serum free thiol levels, which indicate elevated oxidative stress, were associated with a roughly twofold greater risk of 30-day problems and an extended hospital stay in gastrointestinal cancer surgery [33]. These results demonstrate the predictive significance of oxidative biomarkers in perioperative assessment of risk. At the same time, oxidative stress and inflammation are closely related. In a number of abdominal surgeries, higher postoperative levels of MDA (malondialdehyde), IL 1, and IL 6 have been linked to higher rates of complications and worse recovery, highlighting a close connection between redox imbalance and widespread inflammation. Higher perioperative MDA and lower GPX levels in patients with laryngeal cancer anticipated risks and poorer quality life following surgery [34], indicating the biomarkers’ wider applicability.

Protocols for Enhanced Recovery After Surgery (ERAS) provide a way to counteract this loop of stress and inflammation. Comparing ERAS approaches to traditional treatment, studies show considerably reduced levels of procalcitonin and CRP on postoperative days three and four [35]. To lessen surgical stress and boost antioxidant defenses, these procedures use multimodal analgesia, early mobilization, and minimal fasting.

All of these findings point to a distinct mechanism: neuroendocrine stimulation brought on by surgical trauma causes oxidative stress and inflammation, which worsen endothelial dysfunction, coagulopathy, and postoperative sequelae. Crucially, tracking inflammatory and redox biomarkers, including serum free thiols, CRP, IL 6, and MDA, can help predict outcomes and direct perioperative care for patients undergoing abdominal surgery.

The inflammatory chain reaction brought on by surgical trauma is largely dependent on these interactions between cytokines, oxidant mediators, and vascular endothelium. Through their direct contacts and the release of bioactive chemicals, platelets play a crucial role in controlling this response.

The most important oxidative stress and inflammatory mediators seen during abdominal surgery are included in Table 1, along with their postoperative importance and the function platelets play in each pathway.

Platelets have become important immune response regulators in addition to their hemostatic role. In order to recognize microbial components (PAMPs) and endogenous danger signals (DAMPs), they express a variety of pattern recognition receptors, including Toll-like receptors (TLRs), which can activate the immune system even in sterile inflammatory settings like surgery [36].

IL-1β, RANTES, and platelet factor 4 (PF4) are among the immunomodulatory cytokines and chemokines released by active platelets, which alter the influx of leukocytes and enhance stimulation of endothelial cells [37,38]. They contribute to the complex immunological milieu after abdominal surgical trauma by enhancing neutrophil extracellular trap (NET) development and promoting the generation of inflammatory cytokines through their ability to form heterotypic aggregates with neutrophils, monocytes, and lymphocytes [39].

Additionally, by engaging with dendritic cells and T lymphocytes, these platelets can affect the adaptive immune system and contribute to the delivery of antigen through molecules of the MHC class I [40,41]. Because of their diverse functions, platelets may serve as immune sentinels that connect innate and adaptive reactions. This immunologic flexibility is especially important in abdominal surgery because it can lead to both positive outcomes like infection control and negative ones like sepsis or postoperative inflammatory conditions [42,43]. New opportunities for perioperative surveillance and focused antiplatelet therapies in surgical patients may arise from an understanding of platelet-driven immune regulation.

## 4. Oxidative Stress and Platelet Dysfunction

Surgical trauma, especially during major abdominal procedures, sets off a series of metabolic reactions that are typified by an excess of reactive oxygen species (ROS). These compounds, which are mostly produced by NADPH oxidase stimulation and mitochondrial metabolism, can upset the redox equilibrium and cause systemic oxidative stress. This leads to complicated interaction between thrombosis, oxidative damage, and immunological regulation, with platelets acting as both a source and a target of ROS [31,44].

ROS are strong second messengers in platelets that increase aggregation and intracellular signaling. In particular, oxidative stress can raise cytosolic calcium levels, activate the phospholipase C and protein kinase C pathways, and cause conformational changes in GPIIb/IIIa, all of which promote fibrinogen binding and platelet clumping [45,46]. A major contributor to the development of microthrombi and postoperative problems such deep vein thrombosis or compromised microcirculation is this redox-driven hyperreactivity.

Additionally, in the surgical context, platelet antioxidant mechanisms, such as glutathione peroxidase (GPx), catalase, and superoxide dismutase (SOD), are often overloaded. Increased oxidative stress platelet disorder and widespread inflammation have all been related to decreased activity of these types of enzymes [47,48].

Antioxidant support has been shown in recent experiments to potentially reduce platelet activation. For instance, lower plasma levels of soluble CD40 ligand (sCD40L), a crucial indicator of stimulation of platelets, and decreased ROS production were linked to the perioperative delivery of L-carnitine in high-risk individuals having a major operation [49].

All of these results point to the critical role that oxidative stress plays in regulating platelet behavior both during and after abdominal surgery. By comprehending this relationship, novel approaches to perioperative risk assessment and focused antioxidant treatments that lower thrombo-inflammatory consequences become possible.

## 5. Platelet Stimulation Clinical Consequences for Abdominal Surgery

Significant clinical ramifications result from the interaction of platelet activation with oxidative stress in the perioperative context, particularly in abdominal surgery. An oxidative microenvironment is created during major surgical procedures when tissue damage, an increased inflammatory response, and temporary ischemia-reperfusion events come together. In addition to changing platelet reactivity, this illness raises the risk of systemic issues such immunological dysfunction, thrombosis, and delayed recovery.

### 5.1. Platelet Reactivity and Thrombotic Risk in Abdominal Surgery

Reactive oxygen species (ROS) are produced as a result of surgical trauma, and this promotes platelet adhesion, activation, and aggregation. This leads to hypercoagulability, which raises the risk of arterial thrombosis, venous thromboembolism (VTE), and microvascular occlusion, especially in patients who have a metabolic syndrome, underlying cancer, or a history of thrombotic events [9,50]. Oxidative stress has been linked to worse wound healing and an increase in early postoperative thrombotic events in colorectal and hepatobiliary surgery.

Furthermore, by altering receptor signaling pathways, ROS may compromise the effectiveness of perioperative antiplatelet medications—a condition known as “antiplatelet resistance” [49]. This emphasizes that patients who are at a high risk of cardiovascular events or surgical complications require customized antithrombotic treatments.

### 5.2. Inflammation and Immunothrombosis: The Platelet–Immune Axis

Oxidative stress causes platelets to release soluble mediators including sCD40L and P-selectin as well as microparticles, which actively interact with immune cells like neutrophils and monocytes. In addition to exacerbating surgical consequences such as organ malfunction, sepsis, or systemic inflammatory response syndrome (SIRS), these interactions intensify systemic inflammation [37]. Delays in wound healing, tissue damage, and compromised host defenses have all been linked to oxidative platelet activation. Patients who are elderly, immunocompromised, or enduring substantial tumor resections or re-operations should be especially concerned about these complications [51].

Inflammation, in turn, can contribute to increased oxidative stress by promoting the production of ROS and impairing antioxidant defenses. This creates a vicious cycle where oxidative stress and inflammation reinforce each other, contributing to the progression of various diseases [52]. Inflammation can activate platelets and promote their aggregation and adhesion to the vessel wall. Activated platelets can also release factors that further amplify inflammation, creating a positive feedback loop [53]. In addition, platelets express Toll-like receptors (TLRs), which can recognize inflammatory signals and trigger platelet activation, further linking inflammation and platelet function [54]. While the precise details of these interactions are still being investigated, the general pathways linking oxidative stress, inflammation, and platelet activation are well supported by evidence, particularly in the context of various disease processes.

### 5.3. Platelet-Derived Biomarkers: Prognostic Roles

A number of biomarkers have surfaced as viable diagnostic and prognostic instruments in the perioperative context, given the pivotal role that oxidative stress plays in regulating platelet behavior:Pro-inflammatory signaling and platelet activation are reflected by P-selectin and the soluble CD40 ligand (sCD40L).Oxidative byproducts such as malondialdehyde (MDA) and isoprostane show lipid peroxidation in platelet membranes [33].NADPH oxidase activity and platelet-derived ROS production are indicated by sNOX2-derived peptide (sNOX2-dp) [55].

In the perioperative setting, assessing platelet function and oxidative stress markers can improve risk stratification and personalize surgical therapy for abdominal surgery. By measuring platelet reactivity and oxidative stress levels, clinicians can identify patients at higher risk of bleeding or thrombosis, allowing for tailored interventions such as adjusting antiplatelet therapy or optimizing antioxidant strategies. Platelet function tests, such as light transmission aggregometry or flow cytometry, can identify patients with excessive or insufficient platelet reactivity. These procedures can help in distinguishing between individuals who are predisposed to bleeding and those who are susceptible to thrombosis [56,57].

Clinical professionals can use these biomarkers to stratify thrombo-inflammatory risk, track oxidative burden, and inform choices about antiplatelet or antioxidant therapy throughout the perioperative phase.

Measuring sCD40L and sNOX2-derived peptide levels in cancer patients before surgery can help identify individuals at higher risk of perioperative complications, including thrombosis, bleeding, infection, and prolonged recovery. Increased levels of certain platelet activation markers (such as P-selectin and CD40L) are associated with disease progression or cancer-associated venous thromboembolism. This molecule, shed from activated platelets and other cells, is involved in immune and inflammatory responses. Elevated sCD40L levels have been linked to cancer development, progression, and metastasis. In the perioperative setting, high sCD40L levels may be associated with an increased risk of thrombosis, inflammation, and other complications [58,59]. Furthermore, cancer cells can transfer RNA, including mRNA, circRNA, and lncRNA, into platelets, creating “tumor-educated platelets”. These RNA profiles can potentially be used to identify the presence of cancer and even predict the type of cancer [60]. Thrombocytosis is also observed in many cancers and is linked to poorer survival rates (in various cancers, including colorectal, lung, and ovarian) and increased risk of metastasis [61]. Conversely, low platelet counts (thrombocytopenia) can result from cancer treatments like chemotherapy or from cancer itself, particularly in hematologic malignancies, especially leukemia, lymphoma, myelodysplastic syndromes, and multiple myeloma, due to direct infiltration of the bone marrow, impairing platelet production. While less common than in hematologic cancers, solid tumors can also cause thrombocytopenia, either through bone marrow involvement or as a paraneoplastic syndrome [62].

Metabolic syndrome, characterized by a cluster of conditions like obesity, high blood pressure, and insulin resistance, is linked to increased cardiovascular risk and poorer surgical outcomes. In metabolic syndrome, platelets can become hyperactive and contribute to the progression of atherosclerosis and thrombotic events. Elevated sCD40L levels have been associated with increased risk of cardiovascular events in various clinical settings, including acute coronary syndromes and after hip and knee arthroplasty. Measuring sCD40L levels before surgery could help identify patients with metabolic syndrome who are at higher risk of perioperative complications such as myocardial infarction, stroke, or bleeding [63].

Furthermore, in elderly patients, several biomarkers can indicate platelet activation, a process potentially linked to increased cardiovascular risk. These markers include platelet factor 4, P-selectin, and CD40 ligand [64].

However, further research is needed to standardize the measurement of these biomarkers and validate their clinical utility in diverse populations.

### 5.4. Antioxidants and Targeted Modulation in Personalized Therapies

Research is being conducted on therapeutic approaches to lower oxidative platelet activation. In modest clinical and experimental research, agents like vitamin C, N-acetylcysteine, and L-carnitine have demonstrated promise in reducing ROS generation and enhancing platelet function [49]. Despite promising results, more investigation is required to establish the best time, dosage, and patient selection criteria as well as to validate these findings in larger, multicenter trials. Incorporating platelet redox indicators into standard perioperative evaluation may eventually aid in the creation of individualized surgical care plans, improving recovery and safety. The role of antioxidant therapies like L-carnitine, N-acetyl cysteine, and vitamin C in abdominal surgery is not yet fully established and integrated within the classical clinical evidence framework [65].

## 6. Antiplatelet Techniques and Treatment Aspects

The management of platelet activity in the perioperative period, particularly in patients undergoing abdominal surgery, poses a significant clinical challenge. Balancing the risks of thrombosis and bleeding is essential, especially in patients with cardiovascular comorbidities, malignancies, or chronic inflammatory conditions. Oxidative stress further complicates this balance by enhancing platelet reactivity and potentially reducing the effectiveness of standard antiplatelet therapies.

### 6.1. Antiplatelet Agent Use in Patients Having Surgery

In order to avoid arterial thrombosis and cardiovascular events, antiplatelet medications, including aspirin and P2Y12 inhibitors (such as ticagrelor and clopidogrel), are frequently utilized. Individual evaluation is necessary for their perioperative use in abdominal surgery. According to international standards, antiplatelet medication may be necessary in high-risk individuals, notwithstanding the risk of surgical bleeding, such as those with peripheral artery disease or recent coronary stenting [66].

However, interim discontinuance or bridging measures are frequently taken into consideration for major abdominal procedures that have a high risk of bleeding (such as pancreatic or hepatic resections). In order to reduce hemorrhagic risk and limit thrombotic consequences, multidisciplinary planning is crucial. Clinical practice employs a broad range of antiplatelet medications, each with unique pharmacologic characteristics and perioperative consequences. A comparative overview of the most widely used drugs is given in Table 2, together with information on their mechanism of action, bleeding risk profile, and current surgical care guidelines. For patients undergoing abdominal surgery, particularly those with cardiovascular problems, the timing of bridging or termination is crucial in balancing the risks of thrombosis and hemorrhage.

### 6.2. Risks and Benefits of Perioperative Antiplatelet Therapy

Antiplatelet therapy reduces the risk of thromboembolic events, including myocardial infarction, venous thrombosis, and pulmonary embolism, yet it also increases the likelihood of perioperative bleeding, which may lead to blood transfusions, surgical reinterventions, or prolonged hospitalization [33].

Additionally, oxidative stress induced by surgical trauma can reduce the efficacy of standard antiplatelet drugs by altering platelet receptor signaling and promoting resistance—particularly in patients with cardiovascular disease, diabetes, or cancer [49]. This phenomenon, known as oxidative stress-induced antiplatelet resistance, has become increasingly relevant in high-risk surgical populations.

The optimal perioperative antiplatelet strategy balances the risk of thromboembolic complications from stopping antiplatelet therapy against the risk of bleeding from extensive abdominal surgery. Aspirin is a cornerstone of antiplatelet therapy and is often continued perioperatively, especially in low to moderate bleeding risk procedures. Thienopyridines (e.g., clopidogrel, prasugrel, ticagrelor), which inhibit the P2Y12 receptor, are often discontinued 5–7 days before surgery due to their longer duration of action and higher bleeding risk [67].

A particular situation in which the decision to discontinue antiplatelet therapy may be difficult to make is in patients who have recently had a stent placed in a specific vessel. In these cases, the risk of stent thrombosis in the context of premature discontinuation of antiplatelet therapy must be considered. At the same time, the high bleeding risk of continuing antiplatelet therapy in patients undergoing surgery must also be taken into account. Current practice guidelines established by the American Society of Anesthesiologists advocate for the deferral of non-cardiac surgical procedures for a minimum period of 12 months, pending the completion of endothelization of the drug-eluting stent (DES). The 2016 American College of Cardiology and the American Heart Association (ACC/AHA) recommend a waiting period of six months for stable coronary artery disease [68].

### 6.3. Emerging Directions: Selective Inhibitors, Biomarkers, Personalized Strategies

Recent advances in platelet biology and redox medicine have revealed new therapeutic strategies:Selective inhibitors targeting platelet–leukocyte interactions, such as blockers of CD40L or P-selectin, have shown promise in dampening thromboinflammatory pathways without impairing primary hemostasis [37].Redox-sensitive biomarkers—including sCD40L, sNOX2-dp, and MDA—may serve as valuable tools for perioperative risk stratification and therapy adjustment, based on individual oxidative burden and platelet activity [55].Personalized antiplatelet therapy is gaining momentum by integrating genetic profiles, comorbidities, and platelet function markers. Adjunctive antioxidant therapies (e.g., L-carnitine, N-acetylcysteine) have shown potential to reduce oxidative platelet activation while preserving hemostasis [49].

These strategies may pave the way toward more precise and individualized perioperative care. Future research should focus on validating these approaches in large clinical cohorts and refining algorithms that incorporate oxidative stress markers into antiplatelet management.

## 7. Future Directions and Research Gaps

Despite increasing recognition of the roles played by platelets, oxidative stress, and inflammation in the perioperative context, there remain substantial gaps in understanding how these factors interact and influence patient outcomes in abdominal surgery. Addressing these gaps through focused translational research and clinical trials is crucial for optimizing perioperative care.

### 7.1. Current Needs in Translational Research

Current research has identified platelet hyperreactivity and oxidative stress as key contributors to adverse perioperative outcomes, including thrombosis, delayed healing, and systemic inflammation. However, the molecular crosstalk between platelets and oxidative pathways remains insufficiently characterized in surgical patients.

There is a pressing need for translational studies that bridge bench and bedside—clarifying how platelet activation markers (e.g., P-selectin, CD40L), oxidative enzymes (e.g., NOX2), and pro-inflammatory mediators (e.g., IL-6, TNF-α) interact in the setting of surgical trauma [37,55]. Such insights could inform targeted therapies that modulate platelet reactivity while minimizing systemic oxidative damage.

Moreover, animal models of surgery-induced oxidative imbalance remain underutilized. These models could provide mechanistic insights into the feedback loops linking oxidative stress, platelet activation, and immune dysregulation.

### 7.2. Relevance of Platelet Biomarkers in Abdominal Surgery

The perioperative monitoring of platelet-derived biomarkers offers a promising tool for risk stratification and therapeutic adjustment. Biomarkers such as soluble CD40L, sNOX2-derived peptides, and thromboxane B2 have been linked to both thrombotic risk and oxidative burden in surgical settings [49].

However, these markers are not yet routinely integrated into clinical workflows. There is a need for prospective validation studies to determine their predictive value in abdominal surgery, particularly for complications such as postoperative thrombosis, infection, or organ dysfunction. Additionally, the standardization of biomarker thresholds and assay methods is necessary to enable broader clinical adoption.

### 7.3. Potential for Randomized Clinical Trials and Development of Clinical Guidelines

Most current recommendations regarding perioperative antiplatelet use and oxidative stress modulation are extrapolated from cardiovascular or oncologic populations. Dedicated randomized controlled trials (RCTs) in surgical cohorts are needed to evaluate the following:The impact of maintaining vs. suspending antiplatelet therapy in oxidative stress-prone surgeries;The role of adjunctive antioxidant therapy in modulating platelet function;The effectiveness of biomarker-guided strategies in reducing complications and improving outcomes.

Such trials would not only strengthen the evidence base but also facilitate the development of clinical guidelines specific to surgical disciplines. These guidelines should address patient stratification, biomarker monitoring, and personalized perioperative strategies, ultimately aiming to reduce morbidity and mortality in high-risk surgical patients.

Despite significant advancements in understanding platelet-mediated inflammation and oxidative stress, key knowledge gaps persist—especially in the context of major abdominal surgery. Current research is limited by a lack of robust biomarkers, inconsistent clinical validation, and a shortage of high-quality randomized studies that evaluate platelet-targeted interventions in surgical patients. These limitations hinder the development of personalized treatment pathways and evidence-based perioperative protocols. Table 3 outlines the most pressing areas for future investigation, identifies existing gaps in translational platelet research, and highlights the potential clinical impact of resolving these uncertainties in abdominal surgery populations.

## 8. Clinical Relevance

Platelets, long regarded primarily as mediators of hemostasis, have recently been recognized as critical modulators of both inflammation and oxidative stress, particularly in the context of abdominal surgery. Surgical trauma initiates a complex biological cascade in which platelet activation is significantly enhanced by oxidative stress, contributing to a prothrombotic and proinflammatory environment. This response plays a pivotal role in the development of postoperative complications such as thrombosis, impaired wound healing, and systemic inflammatory reactions [37,69].

The identification of platelet-related biomarkers—such as soluble CD40 ligand (sCD40L), thromboxane B2, and sNOX2-derived peptide—offers new opportunities for perioperative risk stratification and therapeutic decision-making [33,55]. These biomarkers may help clinicians identify patients at elevated risk and tailor treatment plans accordingly.

In parallel, perioperative antiplatelet therapy remains a clinical challenge, requiring careful risk–benefit assessment, especially in patients with concomitant cardiovascular or oncologic conditions. Guidelines recommend a case-by-case evaluation, balancing the risks of thrombotic events against those of perioperative bleeding [66].

Emerging therapeutic strategies include the use of selective platelet inhibitors, antioxidant adjuncts, and biomarker-guided protocols aimed at achieving personalized, risk-adapted management [69,70]. These approaches hold promise for improving outcomes in complex surgical patients.

Integrating platelet-focused diagnostics and therapies into surgical practice could support the advancement of precision medicine in abdominal surgery, reduce complications, and enhance the safety and efficacy of perioperative care.

## 9. Conclusions

The interplay between platelet activity, inflammation, and oxidative stress has emerged as a critical determinant of surgical outcomes, particularly in abdominal procedures. Platelets, traditionally recognized for their hemostatic role, are now understood to be central mediators of immune responses and redox signaling. Upon surgical insult, platelet activation is amplified by oxidative stress, contributing to thromboinflammatory cascades that influence healing, infection risk, and thrombotic events. The dynamic crosstalk among platelets, reactive oxygen species, and inflammatory mediators such as IL-6, TNF-α, and CD40L positions platelets as both biomarkers and effectors of perioperative complications [37,55,69].

Understanding these mechanisms has several practical implications for the perioperative management of surgical patients. First, it underscores the importance of individualized antiplatelet strategies, particularly in patients at high risk of cardiovascular or oncologic complications. Second, it supports the use of oxidative stress and platelet biomarkers as potential tools for risk stratification, early detection of complications, and therapeutic guidance. Third, it highlights the potential benefit of adjunctive antioxidant or antiplatelet-modulating therapies, which may mitigate excessive platelet activation while preserving hemostasis [67,70].

In surgical decision-making, these insights emphasize the need for multidisciplinary approaches—incorporating surgical, anesthetic, and hematologic perspectives—to optimize timing, pharmacologic interventions, and postoperative monitoring.

Ultimately, platelets should be regarded not merely as mediators of clot formation but as strategic regulators of the inflammatory and oxidative landscape in abdominal surgery. Their dual role as sensors and amplifiers of surgical stress places them at the center of future translational and clinical research. Harnessing their diagnostic and therapeutic potential may enable the development of personalized perioperative pathways, reduce complications, and improve long-term outcomes in abdominal surgical patients [33].

## Figures and Tables

**Figure 1 ijms-26-07150-f001:**
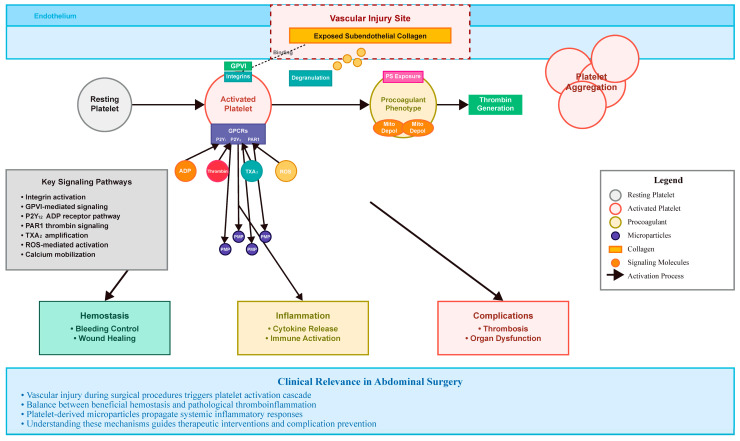
Integrated signal mechanisms and phenotypic transformation of platelets in hemostasis, inflammation, and coagulation in abdominal operations. This figure depicts the main molecular and cellular processes involved in platelet activation after vascular injury. By interacting with exposed subendothelial collagen through integrins and GPVI, resting platelets set off intracellular signaling sequences that result in accumulation, integrin stimulation, and degranulation. Responses to ADP and thrombin are mediated by G-protein-coupled receptors (e.g., P2Y_1_, P2Y_12_, PAR1), whereas activation is amplified by thromboxane A_2_ and reactive oxygen species (ROS). A subset of platelets exhibits a procoagulant phenotype, characterized by exposure to phosphatidylserine and mitochondrial depolarization, which promotes the production of thrombin. Thromboinflammatory signals are further spread by platelet-derived microparticles (PMPs). Both hemostasis and the pathophysiology of postoperative problems depend on these processes.

**Table 1 ijms-26-07150-t001:** Important platelet-mediated and inflammatory mechanisms of postoperative surgical stress: an overview of the roles, mediators, and platelet participation during abdominal surgery.

Mediator/Component	Main Role in Inflammation	Relevance After Surgery	Platelet Participation
IL-6	Pro-inflammatory cytokine	Early postoperative increase, correlates with syndrome severity	Increase in platelet activation and aggregation
TNF-α	Starts an inflammatory chain reaction	Acute phase reaction	Promoting platelet-endothelial adhesion
CRP	Acute-phase reactant	Surge after operation; indicates tissue damage	Indirect platelet stimulation via inflammatory pathways
Endothelial cells	Signaling and barriers	Vascular leakage caused by disruption	Enhanced thrombosis
ROS	Oxidative mediators	Elevated as a result of tissue damage during surgery	Enhanced platelet aggregation and reactivity
Microparticles generated from platelets	Pro-coagulant vesicles	Rise following surgery; associated with complications	Systemic inflammation and platelet activation

**Table 2 ijms-26-07150-t002:** Typical antiplatelet agents in surgical environments: mechanisms, challenges, and pre- and postoperative care.

Antiplatelet Agent	Mechanism of Action	Surgical Bleeding Risk	Perioperative Management Recommendations
Aspirin	Irreversible COX-1 inhibition	Low to Moderate	Often continued for cardiovascular protection; stop 5–7 days prior for high-bleeding-risk surgery
Clopidogrel	Irreversible P2Y12 receptor blocker	Moderate to High	Usually stopped 5–7 days pre-op; consider bridging if high thrombotic risk
Ticagrelor	Reversible P2Y12 inhibitor	High	Discontinued 3–5 days pre-op; short half-life may benefit early resumption
Prasugrel	Irreversible P2Y12 inhibitor	High	Stop 7 days before surgery; associated with higher bleeding risk
Cangrelor	Short-acting reversible P2Y12 inhibitor	Low	Can be used as bridging; rapid offset (1–2 h) advantageous for surgery
Dipyridamole	Phosphodiesterase inhibition, ↑NO	Low	Rarely used perioperatively alone; combined with aspirin in some cases

**Table 3 ijms-26-07150-t003:** Future directions in platelet research in abdominal surgery: key research priorities and potential clinical applications related to platelet involvement in inflammation and oxidative stress.

Research Area	Current Gaps	Potential Clinical Impact
Platelet-derived biomarkers	Limited validation in surgical settings	Risk stratification and early complication detection
Platelet–immune system interactions	Incomplete understanding of postoperative immune modulation	Tailored immunomodulatory therapies
Redox–platelet interplay	Scarce data on dynamic changes perioperatively	Antioxidant therapies guided by platelet activity
Personalized antiplatelet strategies	Lack of predictive markers for bleeding vs. thrombotic risk	Individualized perioperative antiplatelet plans
Randomized controlled trials (RCTs)	Few high-quality studies in abdominal surgery populations	Evidence-based guidelines for platelet-targeted therapies

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
