# Peer review of "Systemic Impact of Platelet Activation in Abdominal Surgery: From Oxidative and Inflammatory Pathways to Postoperative Complications"

_ijms, 2025, doi:10.3390/ijms26157150_

Round 1

Reviewer 1 Report

Comments and Suggestions for Authors

In this manuscript, Scripcariu et al. address the topic “Systemic Impact of Platelet Activation in Abdominal Surgery from Oxidative and Inflammatory Pathways to Postoperative Complications”. The title is succinct and compelling, and the manuscript offers novel and informative contributions to the field. I have outlined several specific comments below that may assist the authors in refining and strengthening their work.

  1. How can platelet function in the perioperative setting and oxidative stress markers be applied to clinical practice for the improvement of risk stratification and individualized surgical therapy in abdominal surgery?
  2. What is the optimal perioperative antiplatelet strategy that weighs preservation of antiplatelet therapy to avoid thromboinflammatory complications against risk of bleeding from extensive abdominal surgery?
  3. Are the findings relating specific platelet parameters (e.g., MPV, PDW, PMP level) to clinical outcomes well supported by current, high-quality evidence?
  4. Is the manuscript sufficiently clear in its characterization of molecular pathways of platelet activation, redox signaling, and receptor-mediated pathways (e.g., GPCRs, CLEC-2, PAR1)?
  5. Are the pathways through which oxidative stress, inflammation, and platelet activation are linked clearly established and supported by available evidence?
  6. Is there sufficient clinical evidence to support the use of specific oxidative or platelet-derived biomarkers (e.g sCD40L, MDA, sNOX2-dp) for risk prediction and outcome prediction?
  7. Are the roles and limitations of antioxidant therapies (e.g., L-carnitine, NAC, vitamin C) appropriately positioned within the classical clinical evidence framework?
  8. Does the article consider patient heterogeneity (e.g., age, cancer status, metabolic syndrome) when it recommends personalized or biomarker-based perioperative strategies?

Author Response

Thank you very much!

Reviewer 2 Report

Comments and Suggestions for Authors

This review by Scripcariu et al explore the evolving role of platelets in abdominal surgery, extending beyond haemostasis to encompass oxidative stress, inflammation, and immune modulation, with recent literature and mechanistic insights with clinical relevance, highlighting platelet-derived biomarkers (ie sCD40L, thromboxane B2, sNOX2-dp) and their prognostic potential. The inclusion of Fig 1 & Table 1-3 summarizing key mediators and antiplatelet strategies enhances clarity and clinical applicability, and outlining future research priorities such as biomarker validation, antioxidant interventions, and RCTs. Overall, the manuscript is well-written and referenced.

Major comments:

1. The review does not explore differences between subpopulations of platelets (ie young vs. senescent platelets), which can vary in reactivity, mitochondrial content, and pro-coagulant activity based on platelet heterogeneity.

2. There is minimal discussion on how platelet production and turnover (ie thrombopoietin regulation, bone marrow response) might influence perioperative platelet function or recovery.

3. The dynamic interaction between activated platelets, leukocytes and endothelial cells is underexplored. For example, ICAM-1, and VCAM-1 mediate interactions with neutrophils, monocytes, and the endothelium - particularly relevant in surgery-induced endothelial dysfunction.

4. To this reviewer, there is a lack of focus on the role of P-selection glycoprotein ligand-1 (PSGL-1) in this manuscript. PSGL-1 on leukocytes and its interaction with platelet P-selectin is central to platelet/leukocyte aggregate formation - a key player in thrombo-inflammation and would be informative to expand under in Section 2, line 74-137.

Minor comment:

While integrins (like αIIbβ3) are mentioned, the manuscript does not explain how bidirectional signalling affects platelet adhesion, spreading, and stable aggregation.

Author Response

Thank you very much!
